# Lactic Acid Bacteria (LAB) and Their Bacteriocins for Applications in Food Safety Against *Listeria monocytogenes*

**DOI:** 10.3390/antibiotics14060572

**Published:** 2025-06-03

**Authors:** Cristian Piras, Alessio Soggiu, Viviana Greco, Pierluigi Aldo Di Ciccio, Luigi Bonizzi, Anna Caterina Procopio, Andrea Urbani, Paola Roncada

**Affiliations:** 1Department of Health Sciences, Magna Græcia University of Catanzaro, 88100 Catanzaro, Italy; c.piras@unicz.it (C.P.); annacaterina.procopio@unicz.it (A.C.P.); 2One Health Unit, Department of Biomedical, Surgical and Dental Sciences, University of Milan, Via Pascal 36, 20133 Milan, Italy; alessio.soggiu@unimi.it (A.S.); luigi.bonizzi@unimi.it (L.B.); 3Department of Basic Biotechnological Sciences, Intensivological and Perioperative Clinics, Università Cattolica del Sacro Cuore, 00168 Rome, Italy; viviana.greco@unicatt.it (V.G.); andrea.urbani@unicatt.it (A.U.); 4Department Unity of Chemistry, Biochemistry and Clinical Molecular Biology, Department of Diagnostic and Laboratory Medicine, Fondazione Policlinico Universitario A. Gemelli IRCCS, 00168 Rome, Italy; 5Department of Veterinary Sciences, University of Turin, Largo Braccini 2, Grugliasco, 10095 Torino, Italy; pierluigialdo.diciccio@unito.it

**Keywords:** lactic acid bacteria, *Listeria monocytogenes*, bacteriocins, proteomics, microbial competition

## Abstract

Background/Objectives: *Listeria monocytogenes* is a major foodborne pathogen responsible for listeriosis, a serious illness with high morbidity and mortality, particularly in vulnerable populations. Its persistence in food processing environments and resistance to conventional preservation methods pose significant food safety challenges. Lactic acid bacteria (LAB) offer a promising natural alternative due to their antimicrobial properties, especially through the production of bacteriocins. This study investigates the competitive interactions between *Lactococcus lactis* and *L. monocytogenes* under co-culture conditions, with a focus on changes in their secretomes to better understand how LAB-derived bacteriocins can help mitigate the Listeria burden. Methods: Proteomic approaches, including Tricine-SDS-PAGE, two-dimensional electrophoresis, and shotgun proteomics, were employed to analyze the molecular adaptations of both species in response to bacterial competition. Results: Our results reveal a significant increase in the secretion of enolase by *L. monocytogenes* when in competition with *L. lactis*, suggesting its role as a stress-responsive moonlighting protein involved in adhesion, immune evasion, and biofilm formation. Concurrently, *L. lactis* exhibited a shift in the production of its bacteriocin, nisin, favoring the expression of Nisin Z—a variant with improved solubility and diffusion properties. This differential regulation indicates that bacteriocin production is modulated by bacterial competition, likely as a defensive response to the presence of pathogens. Conclusions: These findings highlight the dynamic interplay between LAB and *L. monocytogenes*, underscoring the potential of LAB-derived bacteriocins as natural biopreservatives. Understanding the molecular mechanisms underlying microbial competition could enhance food safety strategies, particularly in dairy products, by reducing reliance on chemical preservatives and mitigating the risk of *L. monocytogenes* contamination.

## 1. Introduction

Listeriosis is a significant public health concern due to its high mortality rate, which ranges from 20% to 30% in reported cases and can reach 45% among immunocompromised patients or the elderly. *L. monocytogenes* is an opportunistic pathogen capable of surviving extreme conditions such as refrigeration temperatures, high salt concentrations, and acidic environments [1,2]. Its presence in dairy products, particularly in raw milk and soft cheeses, has been frequently linked to foodborne outbreaks. In 2022, listeriosis reached its highest recorded case numbers in the EU/EEA, with notable increases in Austria, Denmark, France, and Hungary. Whole genome sequencing (WGS) has enhanced outbreak surveillance, revealing persistent microbiological clusters across countries (European Centre for Disease Prevention and Control. Listeriosis. In: ECDC. Annual Epidemiological Report for 2022. Stockholm: ECDC; 2024). In 2022, listeriosis was the fifth most reported zoonosis in the EU, with 2738 cases—mainly affecting those over 64. EFSA and ECDC highlighted a 15.9% rise in notification rate compared to 2021 in their latest One Health Zoonoses report [3]. The persistence of *L. monocytogenes* in ready-to-eat foods, particularly in fish products and dairy items, highlights the challenge of controlling this pathogen within the food industry [4]. Listeriosis, though its occurrence is globally rare, poses a severe public health threat due to its high mortality rate and increasing antibiotic resistance, particularly in ready-to-eat foods. The South African outbreak, with 216 deaths, highlights the urgent need for continuous surveillance and control of this resilient foodborne pathogen [5].

The ability of *L. monocytogenes* to form biofilms further complicates its eradication from food processing environments, as biofilms enhance its resistance to disinfectants and other antimicrobial interventions [6,7]. The pathogen is known for its two primary infection pathways: a non-invasive form, characterized by gastrointestinal symptoms such as diarrhea and vomiting, and an invasive form that leads to severe complications, including meningitis, septicemia, and neonatal infections. In pregnant women, listeriosis often presents as an asymptomatic or flu-like illness, yet it can lead to miscarriage, premature birth, or neonatal sepsis [8,9]. Transmission to the fetus occurs via the placenta, causing life-threatening conditions such as granulomatosis infantiseptica. The variability in infection severity is influenced by factors such as infective dose, pathogenicity of the bacterial strain, and the immune status of the host [9,10].

Dairy products are particularly susceptible to *L. monocytogenes* contamination due to the pathogen’s ability to persist in raw milk, cheese, and dairy processing environments [11]. Studies have shown that cows with mastitis can shed *L. monocytogenes* into milk at concentrations ranging from 10^4^ to 10^5^ CFU/mL. Although pasteurization effectively reduces bacterial loads, post-processing contamination remains a critical concern, especially in cheese manufacturing, where environmental sources such as water, equipment, and surfaces contribute to microbial persistence. Soft and semi-soft cheeses, including Camembert, Gorgonzola, and Taleggio, provide favorable conditions for *L. monocytogenes* growth due to their moisture content and pH [12]. The pathogen has been shown to survive cheese ripening processes, with microbial loads remaining detectable even after extended maturation periods. Furthermore, its psychrotrophic nature allows it to proliferate at refrigeration temperatures, emphasizing the need for additional food safety interventions beyond conventional thermal treatments [13].

Lactic acid bacteria (LAB) are a diverse group of Gram-positive, non-spore-forming microorganisms widely recognized for their crucial role in food fermentation and preservation. Their ability to produce organic acids, hydrogen peroxide, and bacteriocins grants them potent antimicrobial properties, which have been extensively studied as natural alternatives to chemical preservatives. The application of LAB in food safety is particularly relevant in controlling *L. monocytogenes*, a resilient foodborne pathogen responsible for listeriosis, a severe infection that primarily affects immunocompromised individuals, pregnant women, newborns, and the elderly [14,15].

Bacteriocins, ribosomally synthesized antimicrobial peptides produced by LAB, have emerged as promising biopreservative agents against *L. monocytogenes* [16,17]. These peptides exhibit potent antimicrobial activity against closely related or pathogenic bacteria, primarily by disrupting bacterial cell membranes. Among LAB-derived bacteriocins, nisin, produced by *Lactococcus lactis*, is the most extensively studied and commercially applied. Nisin functions through a dual mechanism, binding to lipid II, a key component in bacterial cell wall synthesis, while simultaneously forming membrane pores, leading to bacterial cell death. It has been approved as a food preservative (E234) and is widely used in the dairy industry to inhibit the growth of *L. monocytogenes* and other spoilage microorganisms [7,18].

The effectiveness of bacteriocins in food preservation is influenced by several factors, including bacterial competition, environmental conditions, and quorum-sensing mechanisms that regulate their production. LAB strains vary in their bacteriocin production capacity, and studies have demonstrated that specific conditions, such as co-culture with pathogens, can enhance bacteriocin expression. Nisin exists in two natural variants, Nisin A and Nisin Z, which differ by a single amino acid substitution. This variation affects solubility and diffusion properties, with Nisin Z exhibiting improved performance in food matrices. The interplay between LAB and *L. monocytogenes* in competitive environments remains an area of active research, particularly in understanding how bacteriocin production is modulated in response to pathogen presence [19,20].

This study aims to investigate the antagonistic interaction between *L. lactis* and *L. monocytogenes*, focusing on the differential expression of bacteriocins and bacterial proteins under co-culture conditions. Proteomic techniques, including Tricine-SDS-PAGE, one-dimensional electrophoresis, two-dimensional gel electrophoresis, and shotgun proteomics, were employed to characterize the secretomes of the experimental groups. By identifying molecular changes that occur during bacterial competition, this research provides valuable insights into the potential of LAB-derived bacteriocins for controlling *L. monocytogenes* in food environments. The findings contribute to the development of improved biocontrol strategies in the dairy industry, enhancing food safety while reducing reliance on chemical preservatives.

## 2. Results

The samples were analyzed using Tricine-SDS-PAGE to evaluate the presence of peptides with a molecular weight below 20 kDa, which are difficult to quantify using SDS-PAGE. The results obtained, shown in Figure 1a, indicate that the most significant differences among the various experimental groups correspond to proteins with a molecular weight above 6 kDa.

As shown in Figure 1, the most significant differences among the experimental groups are observed in proteins with a molecular weight above 21 kDa. For this reason, the proteins from the different experimental groups were further analyzed using SDS-PAGE. The obtained data indicate the presence of protein bands that were different during bacterial competition, as in lane LAB + LM in Figure 1b (arrows 1 and 2). Considering this variability, the samples were subsequently analyzed using two-dimensional electrophoresis to facilitate protein identification. Using two-dimensional electrophoresis, maps of the cell filtrates from the various experimental groups were generated. Figure 2 shows the two-dimensional maps of the cell filtrates from the three different experimental groups. Figure 2a represents the analysis of the broth only, Figure 2b shows the cell filtrate of *L. lactis*, Figure 2c represents *L. monocytogenes*, and Figure 2d represents the co-culture of both bacteria (the images of the other replicates are shown in Appendix A). Image analysis revealed, as indicated by the arrow in Figure 2a, a protein spot that was more expressed by one of the two bacteria only under co-culture conditions. The protein was excised from the gel and analyzed using MALDI-TOF mass spectrometry, revealing the identification of enolase (MASCOT score 136, sequence coverage 45%) from *L. monocytogenes* (the spectrum is shown in Appendix A).

To better characterize each secretome, a shotgun proteomics analysis was performed using a nano UPLC-MS system (Waters). The peptide digests, derived from the proteins extracted from each medium, were separated using chromatography coupled with mass spectrometry and analyzed using PLGS (ProteinLynx Global Server, version 3.2.0) software. This method, with the use of an internal standard at a known concentration, enabled the identification of the proteins secreted by each medium and their quantification, allowing for a differential analysis. The media from the first experimental group were analyzed, and shotgun expression analysis also confirmed the presence of *L. monocytogenes*’ enolase, which significantly increases following bacterial competition (Figure 3a, Appendix A). Regarding *L. lactis*, we identified a change in the spectrum of nisin, the bacteriocin expressed by the *L. lactis* strain used. Nisin A and Nisin Z are two natural variants of nisin, differing by a single amino acid, and the second one was considerably more expressed in competition conditions (Figure 3b, Appendix A).

The sequencing of nisin retrieved through our experiments produced the results in Table 1. Briefly, the two different peptides are characterized by a single amino acidic substitution in position 50 of the entire chain (Table 1 shows only the peptide sequence without the pro-peptide). The amino acid residue in position 50 of Nisin Z is an asparagin, while *L. lactis* in normal conditions (non-competition with *L. monocitogenes*) produced Nisin A (histidine in position 50).

Structurally, as in the following Figure 4, it is possible to appreciate the difference in the residues in position 50 where the imidazole ring is clearly visible of histidine (Nisin A) that is missing in asparagine of Nisin Z.

## 3. Discussion

Foodborne pathogens, including *L. monocytogenes*, pose a significant public health threat, particularly in vulnerable populations such as pregnant women, infants, the elderly, and immunocompromised individuals. Listeriosis, the disease caused by *L. monocytogenes*, has been linked to severe infections, including meningitis, septicemia, and neonatal listeriosis, often with high mortality rates. Despite stringent food safety regulations, listeriosis outbreaks continue to occur, primarily due to the pathogen’s ability to persist in food processing environments and survive under adverse conditions. This highlights the urgent need for novel and effective biocontrol strategies to mitigate its impact on the food industry [21].

In this study, we investigated microbial competition between *L. lactis* and *L. monocytogenes*, focusing on secretome modifications in response to bacterial interactions. The proteomic analyses revealed significant changes in protein expression profiles under co-culture conditions, demonstrating the dynamic nature of bacterial competition. *L. monocytogenes* showed an upregulation of enolase, a glycolytic enzyme known to be involved in bacterial adhesion, invasion, and metabolic adaptation to stress. The increased expression of enolase in response to competition suggests that *L. monocytogenes* may employ adaptive mechanisms to counteract the antimicrobial activity of *L. lactis* and maintain its survival within competitive microbial ecosystems [22,23].

The findings of this study provide compelling evidence that *L. monocytogenes* enolase plays a significant role in bacterial competition, particularly in response to *L. lactis*. The proteomic analyses revealed that enolase was significantly upregulated under co-culture conditions, suggesting that its secretion is specifically triggered by bacterial interaction rather than being a constitutive process. This observation aligns with previous reports of enolase functioning as a moonlighting protein, serving not only in glycolysis but also as a key factor in bacterial adhesion, invasion, and immune evasion [24]. The two-dimensional electrophoresis analysis confirmed that enolase secretion occurs only when *L. monocytogenes* is in competition with *L. lactis*, implying that bacterial stress or interspecies interactions induce its extracellular expression. This is further reinforced by shotgun proteomics analysis, which quantitatively demonstrated a significant increase in enolase levels under competitive conditions. The selective secretion of enolase may enhance *L. monocytogenes*’ ability to survive and persist in environments where other bacterial species are present, potentially aiding in biofilm formation and host colonization [25]. From a pathogenic perspective, the increased secretion of enolase under competitive conditions suggests that *L. monocytogenes* may employ it as a virulence-associated factor, facilitating host adhesion or immune system modulation. This is particularly relevant in the context of enolase’s known role in binding plasminogen, a mechanism that enhances bacterial dissemination through tissue barriers [24]. The interplay between enolase and nisin, the bacteriocin produced by *L. lactis*, further supports this hypothesis. The proteomics data indicate a shift from Nisin A to Nisin Z under competition, suggesting that *L. lactis* adapts its antimicrobial strategy to counteract *L. monocytogenes*’ stress response. This bacterial interaction could represent a battle for ecological dominance, in which *L. monocytogenes* uses enolase not only as a metabolic enzyme but also as a weapon to counteract environmental pressures.

On the other hand, *L. lactis* showed the differential expression of nisin, a lantibiotic produced, in response to co-culture conditions. Nisin is widely recognized for its antimicrobial properties against Gram-positive bacteria, particularly *L. monocytogenes*, by disrupting bacterial cell membranes through binding to lipid II. Our results revealed that *L. lactis* significantly increased the production of the Nisin Z variant when exposed to *L. monocytogenes*. This suggests that the presence of a competing pathogen acts as a stimulus for enhanced bacteriocin production, potentially as a protective strategy to outcompete *L. monocytogenes* in shared environments [26]. Nisin antimicrobial effects have been widely documented [27]. Bacteriocins such as nisin play a crucial role in microbial competition by mediating antagonistic interactions between bacterial species. Their production is often regulated by quorum sensing, a cell-density-dependent signaling mechanism that allows bacteria to coordinate gene expression in response to population dynamics and environmental stressors [28]. The observed upregulation of Nisin Z in response to *L. monocytogenes* represents a new finding and demonstrates that bacteriocin synthesis can be modulated by interspecies interactions. This suggests that microbial competition can be exploited to enhance biocontrol strategies in food safety applications, particularly in dairy products where *L. monocytogenes* remains a persistent concern.

The presence of *L. monocytogenes* in dairy products, including soft cheeses and raw milk, poses a substantial risk to food safety. This pathogen can survive and grow under refrigeration conditions and is highly resilient to environmental stressors such as salt and acidity [29]. Previous reports have linked listeriosis outbreaks to contaminated dairy products, emphasizing the need for improved preventive measures. Traditional food preservation techniques, such as heat treatments and chemical preservatives, are effective but may impact product quality and consumer acceptance. In contrast, LAB-derived bacteriocins offer a natural and sustainable alternative for enhancing food safety without compromising sensory or nutritional attributes.

Our findings contribute to the growing body of evidence supporting the application of LAB as bioprotective agents in food systems. The ability of *L. lactis* to increase Nisin Z production in response to *L. monocytogenes* provides valuable insights into microbial interactions that could be harnessed to develop targeted biocontrol interventions. By selecting LAB strains with enhanced bacteriocin production capabilities, it may be possible to create functional starter cultures that provide an additional layer of protection against foodborne pathogens. Nisin Z has greater solubility at neutral pH due to the asparagine residue and, as a consequence, better diffusion in agar, often leading to larger inhibition zones in antimicrobial assays [30]. Both variants share the same antimicrobial spectrum and mode of action, targeting lipid II and disrupting bacterial membranes, but Nisin Z’s enhanced physicochemical properties make it more suitable for certain applications, especially in food preservation at neutral pH [31].

Proteomics-based approaches, as utilized in this study, allow for the identification of key molecular changes associated with bacterial competition. By characterizing secretome profiles, it is possible to pinpoint specific proteins and metabolites involved in antimicrobial defense mechanisms. This information can aid in the optimization of food preservation strategies by identifying conditions that promote bacteriocin synthesis while inhibiting pathogen survival. Moreover, proteomic studies can help elucidate the adaptive responses of *L. monocytogenes* to stress, providing valuable knowledge for the development of novel interventions aimed at disrupting its persistence in food environments.

Beyond dairy products, the implications of this study extend to other food sectors where microbial competition plays a role in pathogen control. Fermented foods, for instance, represent an excellent model for studying the antagonistic interactions between LAB and spoilage or pathogenic microorganisms. Understanding how microbial communities regulate bacteriocin production in response to environmental cues can pave the way for innovative biotechnological applications in food preservation. Additionally, the knowledge gained from this study could be applied to the development of bioengineered probiotics with enhanced antimicrobial properties for both food and clinical applications.

## 4. Materials and Methods

### 4.1. Bacterial Strains and Culture Conditions

Strains of *L. lactis* ATCC 11454 and *L. monocytogenes* ATCC 19115, obtained from the American Type Culture Collection (ATCC) and provided by IZSLER, were used in this study. The bacterial strains were stored at −80 °C in brain heart infusion (BHI) broth supplemented with 20% glycerol. Prior to experimentation, both strains were revived and cultured under optimal growth conditions.

For individual cultures, *L. monocytogenes* ATCC 19115 was inoculated in 100 mL of BHI broth at a 1% (*v*/*v*) inoculum concentration and incubated at 37 °C for 24 h under static conditions. Similarly, *L. lactis* ATCC 11454 was inoculated in 100 mL of BHI broth at 1% (*v*/*v*) and incubated at 37 °C for 24 h. For co-culture experiments, *L. monocytogenes* and *L. lactis* were simultaneously inoculated at 0.5% (*v*/*v*) each in 100 mL of BHI broth and incubated at 37 °C for 24 h. An additional co-culture condition was prepared by inoculating *L. monocytogenes* into BHI broth containing *L. lactis* after 5 h of initial growth of the latter. Each experiment was repeated three times, and the growth curves are represented in Appendix A.

### 4.2. Preparation of Supernatants and Lyophilized Pellets

Following incubation, bacterial cultures were centrifuged at 10,000× *g* for 20 min at 4 °C to separate cells from the culture supernatant and possible cellular debris. The collected supernatants were filtered using a 0.45 µm polyvinylidene fluoride (PVDF) membrane to remove residual cellular debris. For protein purification and concentration, proteins from the supernatant were precipitated using a methanol/chloroform/water method. The resulting protein pellet was solubilized in a solution containing 6 M urea and 100 mM Tris-HCl at pH 7.5.

The experimental groups analyzed included BHI control (BHI, 3 replicates), BHI with *L. lactis* (LAB, 3 replicates), BHI with *L. monocytogenes* (LM, 3 replicates), and BHI co-culture of *L. lactis* and *L. monocytogenes* (LAB + LM, 3 replicates), where *L. monocytogenes* was inoculated after 5 h of *L. lactis* growth.

### 4.3. Protein Quantification

Protein concentrations were determined using a Bradford assay with spectrophotometric readings taken in triplicate at 595 nm (Bio-Rad Protein Assay, Bio-Rad Laboratories, Inc., Segrate, Italy). A standard curve was prepared using Bovine serum albumin (Bio-Rad Protein Assay, Bio-Rad Laboratories, Inc., Segrate, Italy) dilutions, covering a linear detection range of 1.2–10.0 µg/mL. Briefly, from 1 to 10 µL of BSA standard (2 µg/µL) and from 1 to 10 µL of each sample to measure were added to 800 µL of water previously pipetted into clean, dry test tubes allowing, in case of the samples, the dilution of UREA (Merck KGaA, Darmstadt, Germany) and other buffer components. Afterward, to each tube, 200 µL of dye reagent concentrate was added, and the mixture was vortexed thoroughly.

Samples were incubated at room temperature for at least 5 min to allow color development. As the absorbance of the assay increased over time, all samples were measured within 1 h of reagent addition. The absorbance was recorded at 595 nm using a spectrophotometer, and protein concentrations were determined by interpolation from the BSA standard curve.

### 4.4. SDS-PAGE Electrophoresis

One-dimensional SDS-PAGE was performed to analyze protein profiles, optimizing separation for molecular weights between 200 and 15 kDa. Protein samples were prepared with Laemmli sample buffer and resolved on 12% acrylamide gels, followed by Coomassie Brilliant Blue (Merck KGaA, Darmstadt, Germany) staining. One-dimensional Tricine-SDS-PAGE was used to analyze protein and peptide profiles, with an optimal molecular weight range between 100 and 1 kDa as described [32]. Protein samples were prepared with Tricine sample buffer and resolved on 10–16% gradient acrylamide gels, followed by NN-silver stain as described [33].

### 4.5. Two-Dimensional Gel Electrophoresis

Protein samples (200 µg per experimental group) were subjected to two-dimensional electrophoresis. First, proteins were separated by isoelectric focusing (IEF) using an immobilized pH gradient (IPG) strip (pH 3–10, GE Healthcare, Uppsala, Sweden). After equilibration, proteins were resolved in a second dimension using SDS-PAGE with 12% acrylamide gels. Gels were stained with colloidal Coomassie solution for 24 h and scanned using a calibrated densitometer (ImageScanner III, GE Healthcare). The digitized images were used for subsequent analysis.

### 4.6. Image Acquisition and Analysis

Gel images were imported into Progenesis SameSpot v4.5 software (Nonlinear Dynamics, UK) for quality assessment and analysis. Parameters such as saturation and dimensions were checked before processing. Statistical analysis of protein spot intensities was performed using the Progenesis Stats module, which applies a one-way ANOVA to log-normalized volumes. Spots with a *p*-value < 0.05 were considered statistically significant.

### 4.7. MALDI-TOF Mass Spectrometry Analysis

Protein bands from one-dimensional electrophoresis and protein spots from two-dimensional gels were excised and subjected to in-gel digestion using 0.01 mg/mL trypsin at 37 °C overnight. Peptides were desalted using C18 ZipTips (Millipore, Billerica, MA, USA) and co-crystallized with a 0.5 mg/mL α-cyano-4-hydroxycinnamic acid (Merck KGaA, Darmstadt, Germany) solution dissolved in acetonitrile–0.1% (*v*/*v*) trifluoroacetic acid (TFA) in water (1:1).

Spectra were acquired using an Ultraflex III MALDI TOF/TOF mass spectrometer (Bruker-Daltonics, Bremen, Germany) in positive reflectron mode as described by Piras et al. [34]. MS data were processed using FlexAnalysis software v3.3 (Bruker-Daltonics, Bremen, Germany), and protein identifications were analyzed by the peptide mass fingerprint tool of MASCOT (v.2.4.1; www.matrixscience.com) by searching against curated databases specific to *Bacteria* taxonomy restricted to *L. lactis* and *L. monocytogenes*. MS parameters were set as follows: carbamidomethylation of cysteines and oxidation of methionines as fixed and variable modifications, respectively; one missed cleavage site was set for trypsin; and maximal tolerance was established at 70 ppm. For protein identification assignment, only Mascot scores with *p* < 0.05 were considered significant [34].

### 4.8. Protein Digestion

Protein reduction and alkylation were performed by adding 100 mM dithiothreitol (Merck KGaA, Darmstadt, Germany) and incubating for 1 h at 37 °C, followed by 200 mM iodoacetamide (Merck KGaA, Darmstadt, Germany) for 1 h at room temperature in the dark. Proteins were digested with sequencing-grade trypsin (Promega, Madison, WI, USA) at a 1:20 (*w*/*w*) ratio and incubated overnight at 37 °C. The digestion reaction was terminated by adding 0.1% (*v*/*v*) trifluoroacetic acid (TFA).

### 4.9. Chromatography and Mass Spectrometry Analysis

A total of 0.6 µg of digested proteins was injected into a nanoACQUITY UPLC System (Waters Corp., Milford, MA, USA) coupled to a Q-Tof Premier mass spectrometer (Waters Corp., Manchester, UK). Enolase from *Saccharomyces cerevisiae* (ScEnolase) digestion (100 fmol) was added to each sample as an internal standard.

Peptides were first trapped and desalted using a Symmetry C18 5 µm, 180 µm × 20 mm precolumn (Waters Corp.) before separation on a NanoEase BEH C18 1.7 µm, 75 µm × 25 cm nanoscale LC column (Waters Corp.) at 35 °C. Separation was achieved using a gradient of 3–40% acetonitrile in 0.1% formic acid over 150 min, followed by a gradient of 40–90% acetonitrile over 5 min and a 15 min rinse at 90% acetonitrile.

The Q-Tof Premier mass spectrometer operated in “Expression Mode”, alternating between low-energy (4 eV) and high-energy (15–40 eV) collision conditions, with a scan time of 0.8 s over the 50–1990 *m*/*z* mass range. Each sample was analyzed in three biological and technical replicates.

### 4.10. Database Search and Protein Identification

LC-MS/MS data were processed using ProteinLynx Global Server (PLGS) v3.0.2 (Waters Corp.). Protein identification was performed using the embedded ion accounting algorithm of PLGS, searching against *L. lactis*, *L. monocytogenes*, or a combined L. + *L. monocytogenes* database (UniProtKB/Swiss-Prot Protein Knowledgebase) with the sequence of ScEnolase (P00924) appended as an internal reference. All the databases were downloaded from uniport (https://www.uniprot.org/, 10 February 2025) in the FASTA format and filtered for taxonomy.

The search parameters included automatic tolerance for precursor and product ions, a minimum of three fragment ions matched per peptide, a minimum of seven fragment ions matched per protein, and a minimum of two peptides matched per protein. Additionally, one missed cleavage was allowed, carbamidomethylation of cysteines was set as a fixed modification, and oxidation of methionines was considered a variable modification. The false positive rate (FPR) threshold was set at <1%.

### 4.11. Protein Expression Profiling

For protein expression profiling, quantitative protein analysis was performed using PLGS, with normalization against the *Saccharomyces cerevisiae* enolase (P00924) internal standard (100 fmol) [34]. Identified proteins were screened based on the following criteria: detection in at least two out of three technical replicates, a significance threshold of *p* < 0.05 or *p* > 0.95, and an expression level ratio of ±0.30 on a natural log scale.

### 4.12. Validation of Experimental Conditions

The same experimental conditions were replicated in milk-based culture media to evaluate protein expression in a food-relevant matrix. *L. monocytogenes* and *L. lactis* were cultured individually and in co-culture in milk. Samples were depleted of milk proteins and subjected to proteomic analysis.

Label-free shotgun proteomics was performed using a nano UPLC-MS system (Waters), with 0.6 µg of each sample separated by liquid chromatography and analyzed via mass spectrometry. Biological and technical triplicates were analyzed, and data were processed using PLGS v3.0.2 (Waters).

## 5. Conclusions

In conclusion, this study highlights the dynamic interplay between *L. lactis* and *L. monocytogenes*, demonstrating the potential of LAB-derived bacteriocins as effective biopreservative agents. The observed increase in Nisin Z production in response to microbial competition underscores the importance of interspecies interactions in modulating antimicrobial activity. Notably, the switch from Nisin A to Nisin Z, driven by a single amino acid substitution, enhances solubility and diffusion, potentially improving efficacy in food systems with neutral pH. Additionally, the detection of enolase synthesized by *L. monocytogenes*—a moonlighting protein with known roles in adhesion and virulence—suggests that stress responses may not only affect survival but also influence pathogenicity during microbial interactions. These findings support the continued exploration of LAB-based strategies for food safety, aiming to reduce the burden of foodborne pathogens and ensure the production of safe, high-quality food products. However, limitations remain in understanding how these molecular and ecological dynamics translate to complex food matrices and large-scale applications. Future research should focus on elucidating the regulatory mechanisms underlying bacteriocin synthesis using heat-killed *L. monocytogenes* in culture with alive *L. lactis* cultures.

## Figures and Tables

**Figure 1 antibiotics-14-00572-f001:**
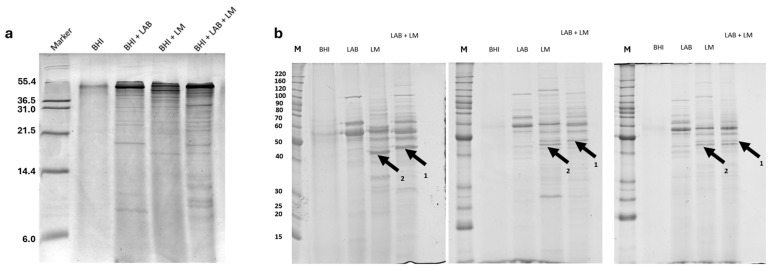
(**a**) Tricine-SDS-PAGE analysis of low molecular weight proteins. (**b**) One-dimensional electrophoresis of the three experimental replicates for each experimental group. MALDI-TOF analysis on bands indicated by arrows 1 and 2 highlighted the presence of enolase. Legend for both figures: M: molecular weight marker; BHI: brain heart infusion broth; LAB: lactic acid bacteria; LM: *L. monocytogenes*.

**Figure 2 antibiotics-14-00572-f002:**
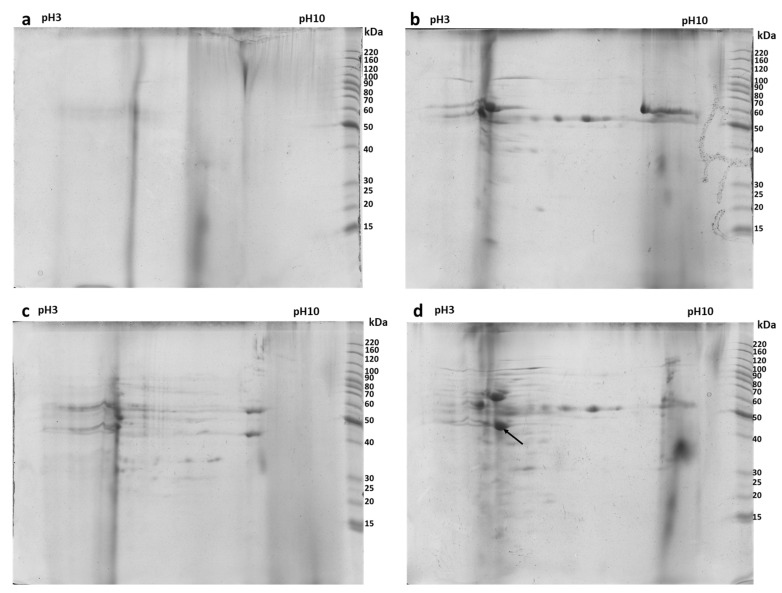
Two-dimensional maps of the cell filtrates from the three different experimental groups: (**a**) BHI medium with no bacterial growth; (**b**) cell filtrate of *L. lactis*; (**c**) cell filtrate of *L. monocytogenes*; and (**d**) cell filtrate of the co-culture of both bacteria. Mass spectrometry analysis identified the protein indicated by the arrow, which is expressed only in the co-culture condition, as the bacterial enolase of *L. monocytogenes*.

**Figure 3 antibiotics-14-00572-f003:**
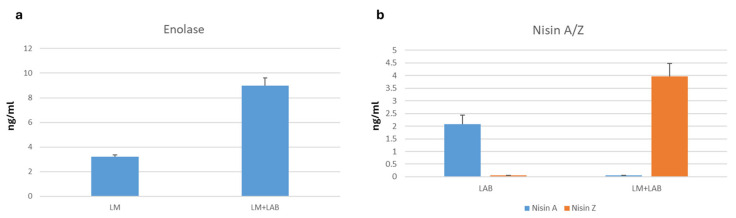
(**a**) Shotgun proteomics analysis results showing the differential representation of Enolase from *L. monocytogenes* alone and in competition conditions expressed in ng/mL. (**b**) Shotgun proteomics analysis results showing the differential representation of nisin lantibiotic forms A and Z in LAB alone and in competition conditions (LM + LAB) expressed in ng/mL.

**Figure 4 antibiotics-14-00572-f004:**
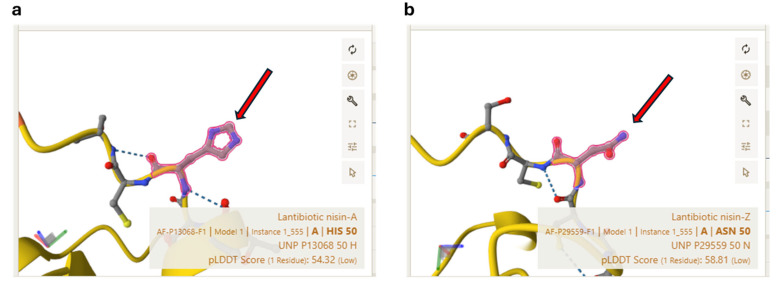
Pairwise structure comparison of Nisin A (**a**) and Nisin Z (**b**) obtained with alphafold (https://alphafold.ebi.ac.uk/entry/P13068 (accessed on 10 May 2025)) starting from PDB models. The red arrows indicate the histidine residue of Nisin A and the asparagine residue of Nisin Z.

**Table 1 antibiotics-14-00572-t001:** Representation of the detected sequence of the lantibiotics Nisin A and Nisin Z. The only different amino acid is represented in bold.

UniProt Accession Number	Description
P13068	Lantibiotic nisin A OS *L. lactis* subsp lactis GN spaN PE 1 SV 1
	ITSISLCTPGCKTGALMGCNMKTATC**H**CSIHVSK
P29559	Lantibiotic nisin Z OS *L. lactis* subsp lactis GN nisZ PE 1 SV 1
	ITSISLCTPGCKTGALMGCNMKTATC**N**CSIHVSK

## Data Availability

The original contributions presented in this study are included in the article/Appendix A. Further inquiries can be directed to the corresponding author.

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
