# Peer review of "Lactic Acid Bacteria (LAB) and Their Bacteriocins for Applications in Food Safety Against Listeria monocytogenes"

_antibiotics, 2025, doi:10.3390/antibiotics14060572_

Round 1
Reviewer 1 Report
Comments and Suggestions for Authors
The study “Lactic acid bacteria (LAB) and their bacteriocins for applications in food safety against Listeria monocytogenes” aims to identify both the bacteriocins produced by LAB and the molecular signals that trigger their secretion.The experimental conditions were well defined but for clarity and reproducibility, I would appreciate additional details regarding the experimental design and data analysis methods. This will help ensure a complete understanding of the study's findings. Below are a few questions and comments to help clarify my understanding and address some uncertainties:
- The introduction is comprehensive but may be too broad in scope.Could the authors consider streamlining it to better focus the narrative and more directly orient readers to the central question of the study?
- Clarification is needed on the protein quantification method.The manuscript states that proteins were resuspended in 6 M urea and quantified using a Bradford assay. Given that high concentrations of urea are known to interfere with Bradford readings, how was this issue mitigated? Were any corrections or alternative quantification methods used?
- Figure 1 legend needs more clarity.Could the authors define all abbreviations used in the figure? Additionally, what do the arrowheads and the labels "1" and "2" indicate? In replicates 2 and 3, band 1 appears similar to the band observed in the LM condition—could the authors elaborate on this?
- The quality of the 2D plots raises concern.The bands appear smeared, which could complicate differential spot identification. Can the authors provide higher-resolution images, and include all replicates for each condition used in protein spot selection?
- More detail is needed on the MALDI analysis.The manuscript mentions that differentially secreted proteins were analysed using MALDI, but does not provide methodological details. Could the authors include parameters or spectra in the supplementary materials?
- Clarification is needed on how absolute protein quantification was achieved.Shotgun proteomics methods, especially label-free approaches, do not typically yield absolute concentration values. How were these values derived, and were any internal standards calibration curves used?
- Reporting of shotgun proteomics data is incomplete.Typically, authors report the number of proteins identified, peptide-spectrum matches (PSMs), fold changes, and p-values for differentially expressed proteins. Additionally, there is no mention of the number of replicates. Could the authors include this information, preferably in a supplementary table?
- The structural data presented are compelling, but the functional relevance could be clarified.Could the authors elaborate on how the structural insights tie into the overall narrative of the study? Specifically, why is the single point mutation highlighted significant in the context of bactericidal activity?
Author Response
The study “Lactic acid bacteria (LAB) and their bacteriocins for applications in food safety against Listeria monocytogenes” aims to identify both the bacteriocins produced by LAB and the molecular signals that trigger their secretion.The experimental conditions were well defined but for clarity and reproducibility, I would appreciate additional details regarding the experimental design and data analysis methods. This will help ensure a complete understanding of the study's findings. Below are a few questions and comments to help clarify my understanding and address some uncertainties:
- The introduction is comprehensive but may be too broad in scope.Could the authors consider streamlining it to better focus the narrative and more directly orient readers to the central question of the study?
Response: Thanks for this comment that helped to make the introduction more appealing to the reader. We focused better, especially in the first part, with the description of the Listeriosis problem.
- Clarification is needed on the protein quantification method.The manuscript states that proteins were resuspended in 6 M urea and quantified using a Bradford assay. Given that high concentrations of urea are known to interfere with Bradford readings, how was this issue mitigated? Were any corrections or alternative quantification methods used?
Response: Thanks, as this part was indeed needing clarification. Now better specified from lines 323 to 337.
- Figure 1 legend needs more clarity.Could the authors define all abbreviations used in the figure? Additionally, what do the arrowheads and the labels "1" and "2" indicate? In replicates 2 and 3, band 1 appears similar to the band observed in the LM condition—could the authors elaborate on this?
Response: The arrows are indicating the areas with major variability needing more investigation (2D electrophoresis). Now better described in lines 137-141.
- The quality of the 2D plots raises concern.The bands appear smeared, which could complicate differential spot identification. Can the authors provide higher-resolution images, and include all replicates for each condition used in protein spot selection?
Response: The images quality is related to the characteristics of the sample (concentration via lyophilization of around 100 mL) where the residual presence of salts remaining from the supernatants negatively interferes with the electrophoresis. The images of the replicates were added in figure 1 SI.
- More detail is needed on the MALDI analysis.The manuscript mentions that differentially secreted proteins were analysed using MALDI, but does not provide methodological details. Could the authors include parameters or spectra in the supplementary materials?
Response: The method was better described in the “MALDI-TOF Mass Spectrometry Analysis” and the requested information has now been added as supplementary material as Figure 2 SI (we can provide the peaklist on request).
- Clarification is needed on how absolute protein quantification was achieved.Shotgun proteomics methods, especially label-free approaches, do not typically yield absolute concentration values. How were these values derived, and were any internal standards calibration curves used?
Response: As internal standard it was used the ScEnolase (P00924) and the quantification was performed according to its peptides’ concentration (lines 4015-420) as described in Piras et al. 2015 (C. Piras et al. / Journal of Proteomics 127 (2015) 365–376)
Reporting of shotgun proteomics data is incomplete.Typically, authors report the number of proteins identified, peptide-spectrum matches (PSMs), fold changes, and p-values for differentially expressed proteins. Additionally, there is no mention of the number of replicates. Could the authors include this information, preferably in a supplementary table?
Response: The number of replicates has now been indicated in the material and methods section (Chromatography and Mass Spectrometry Analysis) and the results describing the proteins quantification of bacterial cultures alone and during competition are indicated in the supplementary information as suggested. In this case, as samples analyzed were the filtrates from different bacterial species/cultures, we thought it was more useful to have a quantitative analysis, rather than a description of the differential expression/representation profiles.
- The structural data presented are compelling, but the functional relevance could be clarified.Could the authors elaborate on how the structural insights tie into the overall narrative of the study? Specifically, why is the single point mutation highlighted significant in the context of bactericidal activity?
Response: Thanks for spotting this lack in the discussion section. A paragraph describing this important point was added between lines 273-278.
Reviewer 2 Report
Comments and Suggestions for Authors
Reviewer Comments
- Title: Please correct Listeria monocitogenes as Listeria monocytogenes
- Introduction: Please include the global disease burden associated with Listeriosis.
- Materials and methods: Please specify if there was any particular reason for using 10,000x g and 30 minutes of centrifugation just to pellet bacterial cells. Generally, 3000-5000g is sufficient as it minimizes potential damage to the cells.
- Table 1: Please italicize Lactococcus lactis.
- Results: During the shotgun proteome analysis of lactis and L. monocytogenes coculture was there an overexpression or underexpression of any other proteins apart from the enolase and Nicin Z?
- Discussion: More details about the nisin A and Z lantibiotics with respect to their functional differences, antimicrobial spectrum and their phylogenetic evolution could be included.
Author Response
- Title: Please correct Listeria monocitogenes as Listeria monocytogenes
Response: Thanks for spotting this mistake, now amended.
- Introduction: Please include the global disease burden associated with Listeriosis.
Response: The listeriosis as global threat has now been discussed in the introduction.
- Materials and methods: Please specify if there was any particular reason for using 10,000x g and 30 minutes of centrifugation just to pellet bacterial cells. Generally, 3000-5000g is sufficient as it minimizes potential damage to the cells.
Response: As the proteomics analysis had to be performed on the secretome, the method was developed to have supernatants as clear as possible without worrying too much about cellular viability. This is the reason for using these parameters in this step. The confusion was generated by a remaining part of the protocol used for bacterial cells wash that has now been removed.
- Table 1: Please italicize Lactococcus lactis.
Response: now done.
- Results: During the shotgun proteome analysis of lactis and L. monocytogenes coculture was there an overexpression or underexpression of any other proteins apart from the enolase and Nicin Z?
Response: The overall results obtained with the shotgun proteomics are now resumed in tables S1, S2 and S3.
- Discussion: More details about the nisin A and Z lantibiotics with respect to their functional differences, antimicrobial spectrum and their phylogenetic evolution could be included.
- Response: Thanks for spotting this lack in the discussion section. A paragraph describing this important point was added between lines 273-278.
Reviewer 3 Report
Comments and Suggestions for Authors
This manuscript addresses an important and timely issue in food microbiology: the use of lactic acid bacteria (LAB) and their bacteriocins, particularly nisin, in controlling Listeria monocytogenes (Lm), a major foodborne pathogen. The work presents novel findings on secretomic and proteomic changes during co-culture of Lactococcus lactis and Lm, highlighting the role of enolase and nisin variants under competitive conditions.
The novelty of the study is somewhat limited by the fact that similar interspecies interactions and the switch in bacteriocin production have been previously suggested in literature. The manuscript should better contextualize what makes this study unique compared to existing findings.
Line 12: Typo in "Departmen Unity of Chemistry" – should be corrected.
Line 177: "is clearly visible the imidazole ring" → "the imidazole ring is clearly visible"
Conclusions could more explicitly state limitations and directions for future research.
The study uses two types of co-culture (simultaneous and delayed inoculation) but only one is analyzed in depth. A rationale for this choice should be explicitly stated.
There is no negative control for potential effects of the medium (BHI) or metabolic byproducts. Control using LAB and Lm in heat-killed or inactivated form would help isolate the impact of active competition.
Author Response
This manuscript addresses an important and timely issue in food microbiology: the use of lactic acid bacteria (LAB) and their bacteriocins, particularly nisin, in controlling Listeria monocytogenes (Lm), a major foodborne pathogen. The work presents novel findings on secretomic and proteomic changes during co-culture of Lactococcus lactis and Lm, highlighting the role of enolase and nisin variants under competitive conditions.
The novelty of the study is somewhat limited by the fact that similar interspecies interactions and the switch in bacteriocin production have been previously suggested in literature. The manuscript should better contextualize what makes this study unique compared to existing findings.
Line 12: Typo in "Departmen Unity of Chemistry" – should be corrected.
Response: Done
Line 177: "is clearly visible the imidazole ring" → "the imidazole ring is clearly visible"
Response: Thanks for the correction, now amended.
Conclusions could more explicitly state limitations and directions for future research.
Response: The discussion was amended accordingly.
The study uses two types of co-culture (simultaneous and delayed inoculation) but only one is analyzed in depth. A rationale for this choice should be explicitly stated.
Response: Now this rationale was better described in lines 120-129 of the introduction section.
There is no negative control for potential effects of the medium (BHI) or metabolic byproducts. Control using LAB and Lm in heat-killed or inactivated form would help isolate the impact of active competition.
Response: Thanks for sharing this suggestion. This will be a part dedicated to the future studies. We included a sentence at the end of the conclusion section.
Reviewer 4 Report
Comments and Suggestions for Authors
The study addresses a timely topic in food microbiology by investigating the proteomic response of Lactococcus lactis and Listeria monocytogenes during co-culture, and highlights the role of bacteriocins, particularly Nisin Z, in the control of L. monocytogenes. The manuscript is generally well-written and well-structured, and the experimental approach is sound. However, several areas require clarification and improvement before the manuscript is suitable for publication.
The Methods section describes two different co-culture conditions. What's the difference between these conditions and specify which one was used for each proteomic analysis. Also, it would be helpful to include bacterial viability data, like growth curves, to show the growth differences between L. lactis and L. monocytogenes in solo culture versus co-culture.
There are also several issues with formatting and consistency. Full species names should be spelled out on first mention and consistently abbreviated afterward. Scientific names should be italicized throughout the text.
In addition, some methodological parts are mentioned but not followed up in the Results. For example, the authors state that lyophilized cultures were saved for analysis, but no corresponding data are shown. Similarly, although co-culture experiments in milk are described, there are no figures or discussion related to this condition. It would be important to either include this data or explain why they were omitted.
Others:
Line 58: The data cited here are over 10 years old. Please consider including more recent data.
Line 125-130: some portions of the figure legends are included within the main body text.
Line 313-314: provide the manufacturer names for the chemicals or products mentioned.
Line 341-345: the “protein quantification” section is duplicated.
Line 368-371: UniprotKB/Swiss-Prot databases was used here, but the release date or version of the database used is not specified. Please include this information to ensure reproducibility and transparency in protein identification.
Author Response
The study addresses a timely topic in food microbiology by investigating the proteomic response of Lactococcus lactis and Listeria monocytogenes during co-culture, and highlights the role of bacteriocins, particularly Nisin Z, in the control of L. monocytogenes. The manuscript is generally well-written and well-structured, and the experimental approach is sound. However, several areas require clarification and improvement before the manuscript is suitable for publication.
The Methods section describes two different co-culture conditions. What's the difference between these conditions and specify which one was used for each proteomic analysis. Also, it would be helpful to include bacterial viability data, like growth curves, to show the growth differences between L. lactis and L. monocytogenes in solo culture versus co-culture.
Response: Thanks for the comment. All the conditions were analyzed via 2D electrophoresis and mass spectrometry, and this concept has now been better clarified through all the manuscript and with the addition of the tables of protein quantification in all the conditions (see tables 1, 2 and 3 in supplementary material). Moreover, the graph indicating the growth curves has been added in supplementary material (Fig. 3 SI).
There are also several issues with formatting and consistency. Full species names should be spelled out on first mention and consistently abbreviated afterward. Scientific names should be italicized throughout the text.
Response: Thanks, now amended.
In addition, some methodological parts are mentioned but not followed up in the Results. For example, the authors state that lyophilized cultures were saved for analysis, but no corresponding data are shown. Similarly, although co-culture experiments in milk are described, there are no figures or discussion related to this condition. It would be important to either include this data or explain why they were omitted.
Response: Thanks for spotting this part that was mistakenly left there from a more general methodological part remaining from the first writing. Now it has been removed.
Others:
Line 58: The data cited here are over 10 years old. Please consider including more recent data.
Line 125-130: some portions of the figure legends are included within the main body text.
Line 313-314: provide the manufacturer names for the chemicals or products mentioned.
Line 341-345: the “protein quantification” section is duplicated.
Response: All these comments were addressed.
Line 368-371: UniprotKB/Swiss-Prot databases was used here, but the release date or version of the database used is not specified. Please include this information to ensure reproducibility and transparency in protein identification.
Response: Now better specified in lines 402-409.
Round 2
Reviewer 1 Report
Comments and Suggestions for Authors
I am satisfied with the revisions made to the manuscript "Lactic acid bacteria (LAB) and their bacteriocins for applications in food safety against Listeria monocytogenes." I appreciate the authors for taking the time and effort to thoughtfully address the comments and make the necessary changes. The revised version has improved in both clarity and completeness.
Reviewer 4 Report
Comments and Suggestions for Authors
The authors have addressed all previous concerns. I have no further comments.